# Successful Treatment of Bloodstream Infection due to a KPC-Producing Klebsiella Pneumoniae Resistant to Imipenem/Relebactam in a Hematological Patient

**DOI:** 10.3390/microorganisms10040778

**Published:** 2022-04-05

**Authors:** Paolo Gaibani, Linda Bussini, Stefano Amadesi, Michele Bartoletti, Federica Bovo, Tiziana Lazzarotto, Pierluigi Viale, Simone Ambretti

**Affiliations:** 1Microbiology Unit, IRCCS Azienda Ospedaliero-Universitaria di Bologna, 40138 Bologna, Italy; stefano.amadesi@gmail.com (S.A.); federica.bovo@aosp.bo.it (F.B.); tiziana.lazzarotto@unibo.it (T.L.); simone.ambretti@aosp.bo.it (S.A.); 2Division of Infectious Diseases, IRCCS Azienda Ospedaliero-Universitaria di Bologna, 40138 Bologna, Italy; linda.bussini@gmail.com (L.B.); m.bartoletti@unibo.it (M.B.); pierluigi.viale@unibo.it (P.V.); 3Section of Microbiology, Department of Experimental, Diagnostic and Specialty Medicine, 40138 Bologna, Italy

**Keywords:** imipenem/relebactam, meropenem/vaborbactam, cross-resistance, *bla*
_KPC_
_-3_

## Abstract

Novel carbapenem-β-lactamase inhibitor combination, *imipenem/relebactam* (IMI-REL), has been recently approved for treatment of infections with limited or no alternative treatment options. In this study, we described the emergence of the IMI-REL-resistance in a KPC-producing *Klebsiella pneumoniae* (KPC-Kp) strain collected from a hematological patient with no evidence of prior colonization. Interestingly, IMI-REL-resistance was associated with meropenem/vaborbactam (MER-VAB) cross-resistance but was not associated with cross-resistance to ceftazidime/avibactam (CAZ-AVI). Although treatment with CAZ-AVI and gentamicin completely eradicated the infection due KPC-Kp cross-resistance to IMI-REL and MER-VAB, the patient became colonized subsequently by KPC-Kp strains susceptible to IMI-REL and MER-VAB. Whole-genome sequencing performed by hybrid approach using Illumina and Oxford Nanopore platforms demonstrated that all KPC-Kp strains isolated from hematological patient belonged to the ST512 and were clonally related. Analysis of antimicrobial and porins genes demonstrated that cross-resistance to IMI-REL and MER-VAB was associated with increased *bla*_KPC-3_ copy number and truncated OmpK35 and OmpK36 with GD134-135 insertion. Phylogenetic analysis demonstrated that KPC-Kp cross-resistance to IMI-REL and MER-VAB was clonally related to a KPC-Kp resistant to IMI-REL as previously described, demonstrating the spread of this multidrug resistant clone in the hematological unit. In conclusion, the results presented in this study reported the emergence of cross-resistance to MER-VAB and IMI-REL in a KPC-Kp strain isolated from a hematological patient and highlight the potential development and diffusion of new multidrug resistance traits.

## 1. Introduction

Carbapenem-resistant *Enterobacteriales* (CRE) represent a public health problem [1]. Resistance to carbapenems is commonly due to the production of plasmid-encoded carbapenemase, which are often associated with several antimicrobial resistance determinants [2,3]. Treatment of infections due to carbapenemase-producing *Enterobacteriales* (CPE) is a particular concern for clinicians, mainly due to the limited available antimicrobial molecules [4]. On the basis of limited options available, a common strategy to overcome MDR resistance is to combine antimicrobial molecules with minimal in vitro activity. Although different studies suggest the superior of the combination therapy rather than monotherapy, many questions remain regarding which combination to use and the effectiveness of combined antimicrobial therapy [1]. The recent introduction of novel β-lactam/β-lactamase inhibitor combinations (BL-BLICs) partially solved the pressing need to treat CPE infections. However, the emerging report of CPE strains resistant to ceftazidime/avibactam (CAZ-AVI) and meropenem/vaborbactam (MER-VAB) limited the clinical impact of novel BL-BLICs [2,3,4]. Recently, a novel carbapenem-β-lactamase inhibitor combination, imipenem/relebactam (IMI-REL), was approved for the treatment of infections caused by CPE. Relebactam is a non-β-lactam β-lactamase inhibitor of the diazabicyclooctane, with activity against class A and class C β-lactamases, including carbapenemase. Relebactam is able to restore in vitro activity of imipenem-cilastatin against Gram-negative pathogens expressing KPC, AmpCs, and extended-spectrum β-lactamases (ESBLs) [5,6,7]. IMI-REL has been evaluated against *Enterobacteriaceae* in several studies, demonstrating excellent in vitro activity with susceptibility rates above 95% for *Klebsiella pneumoniae, Escherichia coli*, *Citrobacter* spp., and *Enterobacter* spp [8]. However, emerging resistance to these new antimicrobial molecules has been recently reported [9,10,11,12].

Herein, we report the treatment of bacteremia in a hematological patient due to a KPC-producing *Klebsiella pneumoniae* (KPC-Kp) resistant to IMI-REL and MER-VAB and susceptible to CAZ-AVI acquired during hospitalization. Antimicrobial treatment with CAZ-AVI and gentamicin rapidly improves clinical condition and selected KPC-Kp strains susceptible to IMI-REL and MER-VAB in rectum.

## 2. Materials and Methods

### 2.1. Bacteria Characterization

Bacteria isolates recovered from the patient’s samples were identified by MALDI-TOF MS assay (Bruker Daltonics, Bremen, Germany). An antimicrobial susceptibility test (AST) was performed using automated system MicroScan Walkaway-96 (Beckman Coulter, Brea, CA, USA). The MICs for IMI-REL, MER-VAB, and CAZ-AVI were determined by using Microtiter Sensititre Plate MDRGN2F (Thermofisher, Waltham, MA, USA) and MIC test strips (Liofilchem, Roseto degli Abruzzi, Italy), as previously described [12,13,14]. MIC values were interpreted following the European Committee for Antimicrobial Susceptibility Testing (EUCAST) breakpoints v11.0 [15]. Carbapenemase production was confirmed by NG-Test CARBA 5 (NG Biotech, Guipry-Messac, France) and/or MALDI-TOF assay and confirmed with molecular assay [9].

### 2.2. Whole Genome Sequencing and Bioinformatic Analysis

During the study period, we collected 1 KPC-Kp strain from blood and 10 KPC-Kp strains from a rectal swab. Whole genomic DNA, extracted from KPC-Kp strains isolated from the patient, was sequenced using Illumina and ONT platforms, as previously described [12]. Briefly, short reads were obtained by using the Illumina iSeq 100 platform (iSeq Reagent Kit v2, Illumina, San Diego, USA) with a 2 by 150 paired-end run after the Nextera DNA Flex paired-end library. Long reads were obtained by using the MinION Mk1C platform (Oxford Nanopore Technologies, Oxford, United Kingdom), with the one-direction (1D) library prepared with a rapid sequencing kit (SQK-RAD004). All read sets were evaluated by FastQC software prior to assembly. Hybrid assemblies were performed with Unicycler version 0.4.7, and contigs were polished with Illumina reads using Pilon v.1.23. Annotation was automatically performed with RAST. 

Assembled genomes were screened for antimicrobial resistance and Sequence type (ST) by the CGE server [16]. ß-lactamase content was manually investigated by BLAST analysis using the Beta-Lactamase-Database [17]. Porin genes and Tn*4401* isoform were manually investigated by BLAST analysis. Phage regions within the KPC-Kp genome were assessed using PHASTER web tool (PHAge Search Tool Enhanced Release) using default parameters [18]. Comparative analysis based on SNPs and indels within the genome’s populations of the KPC-Kp strains was performed by aligning Illumina reads against an annotated genome, as previously described [14]. 

Genetic relationships among isolates included in this study were performed as previously described [11]. Briefly, a phylogenetic tree of the three KPC-Kp genomes was included in this study and, from 58 Italian strains, was generated based on the core genome single nucleotide polymorphisms (SNPs) analysis using Parsnp software and NJST2581 strain as reference. 

Analysis of the *bla*_KPC_ copy number was evaluated using the comparative Ct method by normalizing the *C_T_* value of the target gene to the endogenous 16S rDNA control gene [12]. The fold change (2^–ΔΔC*T*^) was analyzed by using reference strain BAT15.

### 2.3. Genome Accession Number

The sequences of KPC-Kp genomes included in this study were deposited at EMBL/EBI under the project study (EBI project PRJNA809903) with the following accession numbers: BAT15 (SAMN26209579), BAT16 (SAMN26209580), and BAT17 (SAMN26209581).

### 2.4. Ethical Statements

The study was conducted in the context of a normal clinical routine. The study was conducted in accordance with the Declaration of Helsinki. Samples were coded and analysis was performed with an anonymized database.

## 3. Results

### 3.1. Case Description

A 65-year-old man was admitted to the Emergency Department for fever, breathlessness, and fatigue (D0). The patient’s blood cell count showed hyperleukocytosis, anemia, and thrombocytopenia and a peripheral blood smear confirmed the diagnosis of acute myeloid leukemia. Two days later, the patient was admitted to the Hematology Unit. On the same day, cytoreductive therapy and antibiotic treatment with meropenem and tigecycline were introduced. Antibiotic therapy was managed without Infectious Disease (ID) consultation. On D7, chemotherapy with cytarabine, daunorubicin, and midostaurin was started. During the following days, mucositis and febrile neutropenia occurred, and KPC-Kp susceptible to CAZ-AVI, named BAT16, was isolated from blood cultures (D15), while prior surveillance rectal swabs did not show CPE colonization. According to an antimicrobial susceptibility pattern, the patient received a combination therapy with CAZ-AVI and gentamicin. On D21, a rectal swab yielded for once KPC-Kp, named BAT15, though this time the strain was resistant to aminoglycosides while it retained residual susceptibility to colistin and tigecycline. In accordance with active rectal CPE screening in our hospital [19], the patient was monitored weekly for the presence of rectal CPE colonization. Subsequent rectal CPE screening revealed that the patient remained colonized for 4 months by the IMI-REL-susceptible KPC-Kp strain (BAT17). Meanwhile, clinical conditions rapidly deteriorated, and on D23, patient was admitted to the Intensive Care Unit (ICU) for multi-organ failure with septic shock and severe acute kidney injury requiring continuous veno-venous hemodiafiltration. New blood cultures were collected, and after ID consultation, savage antibiotic therapy with high-dose CAZ-AVI (2.5 g every 6h), colistin, and tigecycline was administered. Since blood cultures were negative and considering progressive clinical improvement, this antibiotic regimen was continued up to 14 days. The patient recovered and on D47 was discharged from ICU with residual renal failure needing temporary intermittent dialysis and permanent aminoglycoside-induced bilateral deafness.

### 3.2. Phenotypic Characteristics of KPC-Kp Isolates

Phenotypic characteristics of the KPC-Kp strains included in the study period are shown in Table 1. Antimicrobial susceptibility testing of KPC-Kp strains showed that all strains were resistant to carbapenem (meropenem MICs ≥32 mg/L, imipenem MICs >=32 mg/L) and cefiderocol (MICs range 4-8 mg/L), while all strains were susceptible to CAZ-AVI (MICs range 4-8 mg/L). At the same time, antimicrobial susceptibility analysis revealed that two KPC-Kp strains collected from rectal swabs (BAT15 and BAT17) were susceptible to novel BL-BLICs, while KCP-Kp isolated from blood (BAT16) exhibited an increased MIC of 6 and 2-folds, respectively, for MER-VAB and IMI-REL in comparison to susceptible strains, resulting in resistance to both antimicrobial molecules.

### 3.3. Genome Analysis and Comparison of KPC-Kp Genomes

Genetic characteristics of KPC-Kp strains included in this study are shown in Table 2. Genomic analysis demonstrated that all KPC-Kp belonged to the ST512 and shared similar capsular genes (*wzi-154* and *wzc-916*). Resistome analysis revealed that all KPC-Kp shared similar genes pattern coding for antimicrobial resistance to fluoroquinolone [*oqxA, oqxB*], ß-lactams [*bla_SHV-11,_ bla_TEM1A,_ bla_OXA-9_*], and aminoglycosides [*aac(6’)-Ib, aac(6’)-Ib-cr, aph(3’)-Ia*]. In particular, all strains collected in this study harbored the *bla_KPC-3_* carbapenemase gene. Plasmid content analysis showed that all three KPC-Kp strains carried identical plasmid replicon types [ColRNAI, ncFIB(K), IncFII(K), IncX3, IncFIB(pQil)] and harbored similar phage regions.

Comparison of the core genome SNPs among isolates included in this study showed that BAT15 and BAT17 differed by 38 and 44 SNPs compared to the BAT16 strain. At the same time, alignment of Illumina reads of cross-resistant BAT16 to annotated genomes of IMI-REL susceptible strains (i.e., BAT15 and BAT17) showed that most SNPs occurred within intergenic regions (Appendix A). 

Furthermore, to investigate the basis of cross-resistance to IMI-REL and MER-VAB, we examined the sequence rearrangement of occurred plasmids derived from different strains included in this study. Genome analysis revealed that Tn*4401* was located within 228 Kb [IncFIB(K)]/ IncFII(K)] and 118 Kb [IncFII(K)/ IncFIB(pQil)] plasmids in all KPC-Kp strains, while the third copy present in the IMI-REL and MER-VAB cross-resistant KPC-Kp was found within an IncX3 plasmid of 57 Kb in size. Interestingly, the 57 Kb plasmid carrying the *bla*_KPC-3_ gene in BAT16 showed a backbone common with the IncX3 plasmid from BAT17 and BAT15 strains (Figure 1). The region of difference between IncX3 plasmids was flanked by IS*3000* and IS*26* elements and contained a region harboring the *bla*_SHV-11_ gene in the 43 kb plasmids of BAT15 and BAT17 strains, while it carried a region harboring *bla*_KPC-3_, carrying Tn*4401* transposon in the 57 Kb of IMI-REL-resistant strain (Figure 1). 

Deep sequence analysis demonstrated the region harboring Tn*4401* transposon and, bracketed by IS*3000*–IS*26*, was also present in plasmids of 228 and 118 Kb shared among all KPC-Kp strains, thus resulting in three *bla*_KPC-3_ copies in the IMI-REL-resistant strain and two copies in the IMI-REL-susceptible strains (Figure 2). In order to confirm the relationship between the increased *bla*_KPC_ copy number and IMI-REL resistance, qPCR was performed among all strains included in this study. Relative quantification of the *bla*_KPC-3_ gene demonstrated that IMI-REL-resistant KPC-Kp exhibited an increased copy number rather than IMI-REL-susceptible strains.

During the study period, we isolated a KPC-Kp strain resistant to IMI-REL, MER-VAB, and CAZ-AVI in the same ward from a different patient [20]. Our previous findings demonstrated that KPC-Kp strain cross-resistant to IMI-REL, MER-VAB, and CAZ-AVI, named CAZ47, harbored similar plasmids content and chromosome by exhibiting high sequence identity. Genome comparison demonstrated that BAT16 strain differed to CAZ47 by the presence of different *bla*_KPC_ alleles (*bla*_KPC40_ and *bla*_KPC53_ in CAZ47 and *bla*_KPC3_ in BAT16) and by an additional plasmid of 118 Kb carrying the third copy of *bla*_KPC-3._


In order to confirm the closely genomic relationship of KPC-Kp strains collected from two neutropenic patients, we performed a phylogenetic analysis based on the core genomes SNPs of KPC-Kp genomes isolated in Italy. Our phylogenetic tree showed that BAT15, BAT16, and BAT17 clustered closely with KPC-Kp strains collected from the second patient (i.e., BO318, CAZ59 and CAZ47), thus confirming the clonal spread of KPC-Kp strains cross-resistant to IMI-REL and MER-VAB in the same hematology unit (Figure 3).

## 4. Discussion

In this study, we described the emergence of the cross-resistance to IMI-REL and MER-VAB in a KPC-Kp isolated from a neutropenic patient. Clinical data showed that antimicrobial combination therapy based on CAZ-AVI in combination with gentamicin completely eradicated the infection due to cross-resistant KPC-Kp strain and that, following the BSI episode, the patient became colonized by IMI-REL-susceptible strains, which resulted clonally, related to KPC-Kp cross-resistance to IMI-REL and MER-VAB. Of note, our findings demonstrate that the emerging cross-resistant KPC-Kp was genetically related to a KPC-Kp strain resistance to IMI-REL, MER-VAB, and CAZ-AVI described in our previous study [20]. Based on these findings, it seems that the patient acquired an infection due to cross-resistant KPC-Kp strain during hospitalization and that the combination therapy based on meropenem, in association with tigecycline, did not select strains cross-resistant to IMI-REL or MER-VAB.

In accordance with previous findings [9,20,21], our results confirmed that the IMI-REL-resistance was associated with an increased blaKPC copy number in a OmpK35 porin deficiency strain harboring OmpK36 porin mutations. In particular, previous studies demonstrated that KPC-Kp strains are resistant to IMI-REL exhibited porins deficiency (i.e., truncated OmpK35 and OmpK36 truncated or with GD134-135 insertion) and increased blaKPC copy number (i.e., 2-4 folds) rather than IMI-REL-susceptible strains [20,21]. At the same time, we demonstrated that the mechanism at the basis of IMI-REL-resistance was associated with the cross-resistance to MER-VAB. Genome analysis of KPC-Kp strain cross-resistance to IMI-REL and MER-VAB showed that a region containing blaKPC-3 carrying Tn*4401* transposon was shared among IncX3, IncFIB(pQIL)/IncFII(K), and IncFIB(K)/IncFII(K) plasmids and based on the absence of flank-ing direct repeats. We hypothesized that this region was transposed via homologous recombination mediated by IS3000 and IS26 elements. At the same time, our results demonstrated that the KPC-carrying Tn*4401* transposon was present in triple copies within the cross-resistant KPC-Kp and in double copies among IMI-REL-susceptible KPC-Kp strains included in this study. Similar findings have been reported in previous studies, demonstrating that these elements may be at the basis of the high transferability efficiency of antimicrobial resistance genes in MDR microorganisms [20,21].

In conclusion, here we reported the successful treatment of bacteremia due to KPC-Kp strain resistant both to IMI-REL and MER-VAB. Cross-resistance to IMI-REL and MER-VAB represents an emerging threat that could pose further limitations for clinicians to the treatment of infections due to difficult to treat (DTR) pathogens, especially in critical patients. In light of emergence diffusion of antimicrobial resistance and the recent COVID-19 pandemic, there is an urgent need to optimize antimicrobial therapy and integrate antimicrobial stewardship activities and infection control measures to limit the spread of novel resistance [22]. In this context, further studies should be necessary to identify the suboptimal drug exposure related to the cross-resistance to IMI-REL and MER-VAB, characterize specific-mutations related to cross-resistance emerging in vivo under different antimicrobial therapy, and define the optimal antimicrobial treatment for infections due to cross-resistant KPC-Kp.

## Figures and Tables

**Figure 1 microorganisms-10-00778-f001:**
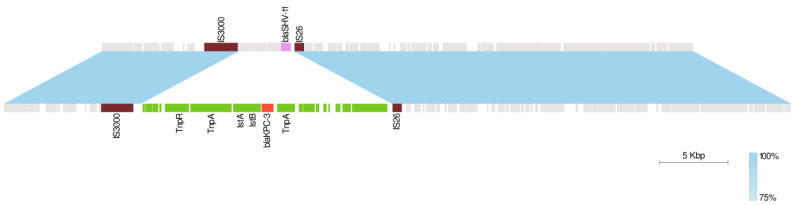
Linear comparison between IncX3 plasmids in BAT15 (**top**) and BAT16 (**bottom**) KPC-Kp isolates. Regions with >75% sequence homology are displayed by a light blue gradient. Boxes indicate annotated coding sequences. The backbone sequence is shown in gray. IS*3000* and IS*26* elements flanking the transposed region are shown in brown. The transposed region is shown in green, *bla*_SHV-11_ gene in pink, and *bla*_KPC-3_ gene in red.

**Figure 2 microorganisms-10-00778-f002:**
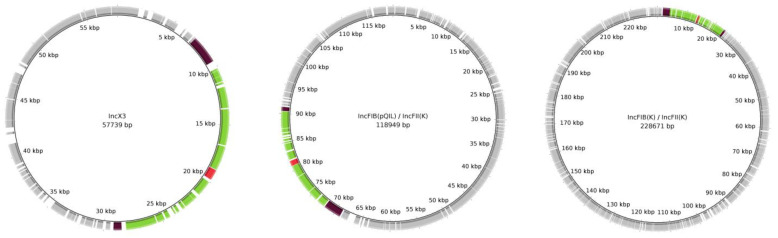
Circular map representing the complete sequence of IncX3, IncFIB(pQIL)/IncFII(K), and IncFIB(K)/IncFII(K) plasmids in BAT16 KPC-producing *Klebsiella pneumoniae* isolate. The backbone sequence is shown in gray. IS*3000* and IS*26* elements flanking the transposed region are shown in brown. The transposed region is shown in green. *bla_KPC-3_* gene is shown in red.

**Figure 3 microorganisms-10-00778-f003:**
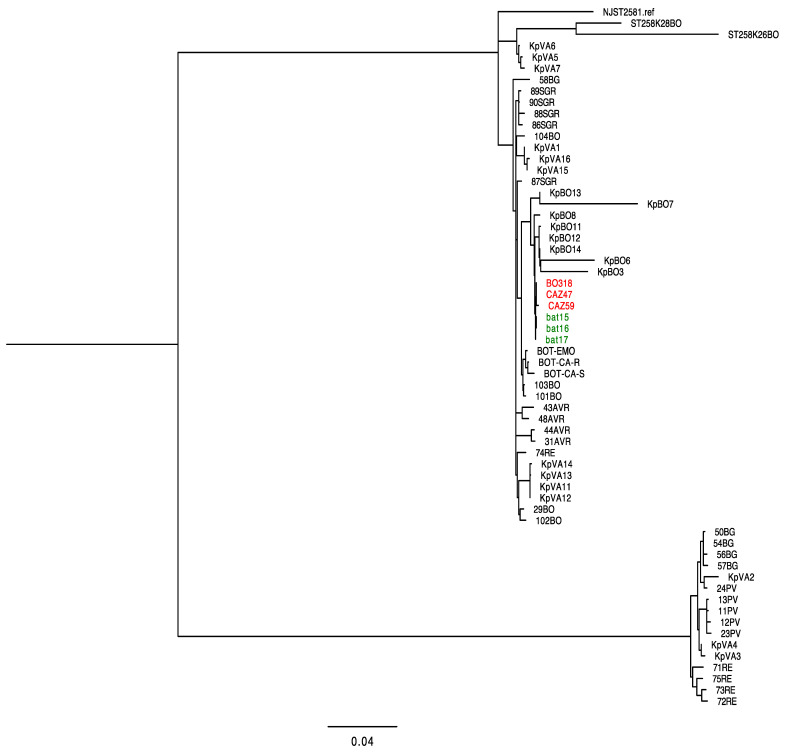
Phylogenetic tree of KPC-producing *K. pneumoniae* (KPC-Kp) included in this study and genomes of Italian isolates. Strains described in this study are highlighted in green, while KPC-Kp strains described in a previous study [20] are highlighted in red.

**Table 1 microorganisms-10-00778-t001:** Phenotypic characteristics of KPC-Kp strains included in this study.

Isolate	Sample	Date of Isolation	MIC (ug/mL)							
			GEN	TGC	IPM	MEM	CAZ-AVI	MEM-VAB	IPM-REL	CFD
BAT15	Rectal Swab	02/13/2020	<2	≤0.5	**>32**	**>32**	4	4	1	8
BAT17	Rectal Swab	03/10/2020	<2	≤0.5	**>32**	**>32**	4	8	2	4
BAT16	Blood	02/06/2020	<2	≤0.5	**>32**	**>32**	8	16	4	8

Abbreviations: GEN, gentamicin; TGC, tigecycline; IPM, imipenem; MEM, meropenem; CAZ-AVI, ceftazidime/avibactam; MER-VAB, meropenem/vaborbactam; IPM-REL, imipenem/relebactam; CFD, cefiderocol. Resistance is shown in bold.

**Table 2 microorganisms-10-00778-t002:** Genotypic traits of KPC-Kp strains included in this study.

Isolate	ST	No. Phage Regions	Capsular Type		Antimicrobial Determinant Genes			Porins			Plasmid	*bla*_KPC_ Copy Number
			*wzi*	*wzc*	Beta-lactams	Fluoroquinolones	Aminoglycosides	*OmpK35*	*OmpK36*	*OmpK37*	Plasmid_replicon (InC) type	
BAT15	512	8	154	916	*bla_KPC-3_*, *bla_SHV-11_*, *bla_TEM1A_, bla_OXA-9_*	*oqxA*, *oqxB*	*aac(6*′*)-Ib*, *aac(6*′*)-Ib-cr*, *aph(3*′*)-Ia*	Truncated at aa 40	GD insertion at aa 134–135	Truncated at aa 229	ColRNAI, ncFIB(K), IncFII(K), IncX3, IncFIB(pQil)	1 *
BAT17	512	8	154	916	*bla_KPC-3_, bla_SHV-11_, bla_TEM1A_*, *bla_OXA-9_*	*oqxA*, *oqxB*	*aac(6*′*)-Ib*, *aac(6*′*)-Ib-cr*, *aph(3*′*)-Ia*	Truncated at aa 40	GD insertion at aa 134–135	Truncated at aa 229	ColRNAI, ncFIB(K), IncFII(K), IncX3, IncFIB(pQil)	1.301
BAT16	512	8	154	916	*bla_KPC-3_*, *bla_SHV-11_*, *bla_TEM1A_*, *bla_OXA-9_*	*oqxA*, *oqxB*	*aac(6*′*)-Ib*, *aac(6*′*)-Ib-cr*, *aph(3*′*)-Ia*	Truncated at aa 40	GD insertion at aa 134–135	Truncated at aa 229	ColRNAI, IncFIB(K), IncFII(K), IncX3, IncFIB(pQil)	1.406

* Used as reference.

## Data Availability

Not applicable.

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
