# Peer review of "Successful Treatment of Bloodstream Infection due to a KPC-Producing Klebsiella Pneumoniae Resistant to Imipenem/Relebactam in a Hematological Patient"

_microorganisms, 2022, doi:10.3390/microorganisms10040778_

Round 1

Reviewer 1 Report

„Successfully treatment of bloodstream infection due to a KPC producing Klebsiella pneumoniae resistant to imipenem- relebactam in a hematological patient”.

Imipenem-relebactam and meropenem-vaborbactam are two novel carbapenem-β-Lactamase inhibitor combinations against KPC-Kp strains.

Authors of the manuscript have been reported Klebsiella pneumoniae resistant to imipenem- relebactam and meropenem/vaborbactam (MER/VAB) using phenotypic characteristics and whole-genome sequencing.   The study included characteristics of KPC-producing Klebsiella pneumoniae strains from one patient in relation to another isolates recovering previously in the same hematology unit confirming the clonal spread of KPC-Kp strains resistant to IMI-REL and MER-VAB.  There are risk clones. 

The Authors wrote an interesting and well written case on the KPC producing Klebsiella pneumoniae resistant to imipenem- relebactam. The manuscript is well structured and includes all relevant information.

Only minor corrections will be needed in order to improve its quality.

Comments and Suggestions for Authors

„Abstract” "bla KPC-3 copy number" „blaKPC-3”  - should be written in italics and subscript similarly to the main body

„Introduction” should be more expanded (regarding the problem of antibiotic resistance Kp)

„Materials and Methods”

How many patient samples, bacterial isolates KPC-Kp have been researched, this information should be included in the materials and methods issue

Ethical Statements – should be included

Version program?

PHAGE web tool -?

databases available from? (please provide versions number and access to databases)

cut-off of identity?

„Results”

Current Fig. 3 is difficult for readers

It will be interesting to see the difference between the cgMLST phylogeny tree and cgSNP phylogeny tree

„Discussion”:

Access to previous studies is difficult (cited work: in press), hence the previous studies should be described in more detail.

The discussion should relate not only to your own research but also to other work related to the topic.

Minor concerns:

Line 75-76: Please rephrase to improve sentence clarity (2x „sequence type”)

„Assembled genomes were screened for Sequence type, antimicrobial resistance, and Sequence type (ST) by CGE server”

Line 38: Imipenem-Relebactam (IMI/REL), change to imipenem/relebactam (IMI-REL)

Line 44:” in vivo” should be written in italics

Line 82: „Genetic relationships among isolates”  - how many isolates were taken to genetic relationships?

Line 87: twice „to”

Line 154: Figure 1. „Klebsiella pneumoniae” -  should be written in italics

Line 163: „bla” - should be written in italics

Line 172 (Figure 2) „IS3000 and IS26 elements” - make it consistent with the main text (according to line 165)

Author Response

Abstract: "bla KPC-3 copy number" “blaKPC-3” - should be written in italics and subscript similarly to the main body „

Authors’ reply and amendments: We thank the reviewer for the insightful comments. We corrected the blaKPC-3 throughout the manuscript as suggested

Introduction” should be more expanded (regarding the problem of antibiotic resistance Kp)

Authors’ reply and amendments: The introduction was expanded by discussing appropriately the problem of antimicrobial resistance and in particular the paucity of antimicrobial molecules active against CPE. 

„Materials and Methods”

How many patient samples, bacterial isolates KPC-Kp have been researched, this information should be included in the materials and methods issue

Authors’ reply and amendments: The number of clinical isolate and total number of KPC-Kp strains collected from rectal swab was added to the Material and Methods section as requested (see line 67)

Ethical Statements – should be included

Authors’ reply and amendments: Ethical Statements was added to the Material and Methods section as requested

Version program?

PHAGE web tool -?

Authors’ reply and amendments: The version of PHAGE web tool (i.e. PHASTER) was added to the Material and Methods section as requested

databases available from? (please provide versions number and access to databases)

cut-off of identity?

Authors’ reply and amendments: The software was used with the default parameters. The sentence was modified accordingly by adding the parameters

„Results”

Current Fig. 3 is difficult for readers

It will be interesting to see the difference between the cgMLST phylogeny tree and cgSNP phylogeny tree

Authors’ reply and amendments: Figure 3 was changed to better shown the tree. Although we agree with referee that could be interest to see the difference between the two analysis approaches, we opted to retain in the manuscript the core genome phylogeny for clarity.  

„Discussion”:

Access to previous studies is difficult (cited work: in press), hence the previous studies should be described in more detail.

Authors’ reply and amendments: The results of previous studies were added to the discussion as requested.

The discussion should relate not only to your own research but also to other work related to the topic.

Authors’ reply and amendments: We agree with referee and we added research by other group describing the resistance to IMI-REL in KPC-Kp strains (see reference 21)

Minor concerns:

Line 75-76: Please rephrase to improve sentence clarity (2x „sequence type”)

Authors’ reply and amendments: The sentence was modified as requested

„Assembled genomes were screened for Sequence type, antimicrobial resistance, and Sequence type (ST) by CGE server”

Authors’ reply and amendments: The sentence was modified as requested

Line 38: Imipenem-Relebactam (IMI/REL), change to imipenem/relebactam (IMI-REL)

Authors’ reply and amendments: The name was changed as requested

Line 44:” in vivo” should be written in italics

Authors’ reply and amendments: the term was modified as requested

Line 82: „Genetic relationships among isolates” - how many isolates were taken to genetic relationships?

Authors’ reply and amendments: The number of isolates used for the analysis was added as requested

Line 87: twice „to”

Authors’ reply and amendments: twice was added as requested

Line 154: Figure 1. „Klebsiella pneumoniae” - should be written in italics

Authors’ reply and amendments: The species name was checked throughout the manuscript

Line 163: „bla” - should be written in italics

Authors’ reply and amendments: The term was modified

Line 172 (Figure 2) „IS3000 and IS26 elements” - make it consistent with the main text (according to line 165)

Authors’ reply and amendments: The name of IS was modified as requested

Reviewer 2 Report

In this work, the authors have reported the successfully treatment of bacteremia due to Klebsiella pneumoniae strain resistant to IMI-REL and MER-VAB. 

It is a new and exhaustive work, but the quality of the figures, especially figure 3, and the discussion must improve to be acceptable for publication.

Author Response

It is a new and exhaustive work, but the quality of the figures, especially figure 3, and the discussion must improve to be acceptable for publication.

Authors’ reply and amendments: We thanks reviewer for his comment. Figure 3 was modified to better shown the tree and the discussion section was revised and enlarged to better discuss the results presented in this study.

Reviewer 3 Report

I read with great interest the paper of Gaibani and colleagues. It is not easy for me give some suggestions because the paper present high quality of research and presentation.

Below only minor consideration: 

  • only minor language check could be considerated
  • in introduction or discussion section add, to contextualize the paper during covid pandemic, that also epidemic has impact on onset of antimicrobial resistance (see and if you want cite Segala FV,  Impact of SARS-CoV-2 Epidemic on Antimicrobial Resistance: A Literature Review. Viruses. 2021 Oct 20;13(11):2110. doi: 10.3390/v13112110.)
  • discuss better the role of IMI-REL-resistance and their determinants
  • conclusion: give some proposal that came from your excellent paper

Author Response

  • only minor language check could be considerated

Authors’ reply and amendments: We thank the reviewer for his comment. The manuscript was carefully checked to correct unclear sentence and typo

  • in introduction or discussion section add, to contextualize the paper during covid pandemic, that also epidemic has impact on onset of antimicrobial resistance (see and if you want cite Segala FV,  Impact of SARS-CoV-2 Epidemic on Antimicrobial Resistance: A Literature Review. Viruses. 2021 Oct 20;13(11):2110. doi: 10.3390/v13112110.)

Authors’ reply and amendments: We added to the discussion section a sentence describing the importance of infection control measures and the importance of monitoring the MDR diffusion especially in COVID-19 pandemic.

  • discuss better the role of IMI-REL-resistance and their determinants

Authors’ reply and amendments: We thank the reviewer for the insightful comment. We discuss better the impact of IMI-REL resistance in the discussion section as requested.

  • conclusion: give some proposal that came from your excellent paper

Authors’ reply and amendments: We thank the reviewer for his comment. We give additional proposal to limit the diffusion of these novel resistances in the discussion section